# MuLan: Adapting Multilingual Diffusion Models for Hundreds of Languages with Negligible Cost

**Sen Xing** [1 2 *]   **Muyan Zhong** [1 *]   **Zeqiang Lai** [3 *]   **Liangchen Li** [2]   **Jiawen Liu** [4]   **Yaohui Wang** [2]   **Jifeng Dai** [1 2]
**Wenhai Wang** [2 3]

## Abstract

In this work, we explore a cost-effective framework for multilingual image generation. We find that, unlike models tuned on high-quality images with multilingual annotations, leveraging text encoders pre-trained on widely available, noisy Internet image-text pairs significantly enhances data efficiency in text-to-image (T2I) generation across multiple languages. Based on this insight, we introduce **MuLan**, **Mu**lti-**Lan**guage adapter, a lightweight language adapter with fewer than 20M parameters, trained alongside a frozen text encoder and image diffusion model. Compared to previous multilingual T2I models, this framework offers: (1) **Cost efficiency.** Using readily accessible English data and off-the-shelf multilingual text encoders minimizes the training cost; (2) **High performance.** Achieving comparable generation capabilities in over 110 languages with CLIP similarity scores nearly matching those in English (39.57 for English vs. 39.61 for other languages); and (3) **Broad applicability.** Seamlessly integrating with compatible community tools like LoRA, LCM, ControlNet, and IP-Adapter, expanding its potential use cases.

## 1. Introduction

Recent diffusion models (Esser et al., 2024; Li et al., 2024a;b; Team, 2024; Wu et al., 2024; Zhang et al., 2022) for content generation have attained stunning advancements in terms of both aesthetic quality and text-content alignment. However, these models still face substantial limitations in multilingual support. For instance, one of the most

*Equal contribution [1]Tsinghua University [2]OpenGVLab, Shanghai AI Laboratory [3]The Chinese University of Hong Kong [4]Johns Hopkins University. Correspondence to: Wenhai Wang <wangwenhai@pjlab.org.cn>.

*Proceedings of the 42$^{nd}$ International Conference on Machine Learning*, Vancouver, Canada. PMLR 267, 2025. Copyright 2025 by the author(s).

popular image-generation models, Stable Diffusion (Rombach et al., 2022), and its successors (Podell et al., 2023; Esser et al., 2024) only supports English and a few Latin-based languages. This language barrier restricts the model's performance in multilingual contexts and hinders its applicability worldwide across diverse cultural and linguistic backgrounds.

The community primarily adopts two approaches to achieve multilingual text-to-image (T2I) generation. (1) The first is *translation-based methods* where input content is temporarily translated into English before generating the image. However, this approach often leads to inference delays, translation errors, and notable issues when handling slang or culturally nuanced content. (2) The other approach is *native multilingual T2I models* (Zhang et al., 2022; Radford et al., 2021; Arkhipkin et al., 2023; Shing & Akiba; Li et al., 2024b; Team, 2024), which is trained directly on high-quality images captioned in the target language. While this native approach improves image generation quality for non-English languages, it relies on extensive, carefully curated image-generation data in the target language, making it data- and resource-intensive. As a result, *exploring a more efficient and generalizable approach to achieve strong multilingual generation capabilities remains challenging.*

On the other hand, thanks to the development of computational power and dataset scale, many existing language models have achieved strong multilingual capabilities through training on large-scale internet data. For example, models trained on text data (*e.g.*, BERT (Devlin et al., 2019), the GPT series (Brown et al., 2020; OpenAI, 2023), and LLaMA (Touvron et al., 2023; Dubey et al., 2024)) and those trained on image-text pairs (*e.g.*, CLIP (Radford et al., 2021), ALIGN (Jia et al., 2021), and InternVL (Chen et al., 2023b)) demonstrate outstanding performance in multilingual understanding. Since the multilingual capability of T2I generation models is closely tied to their text encoders, it becomes essential to explore *how these powerful multilingual text encoders can be leveraged to enable existing generative models to achieve multilingual capabilities more efficiently and effectively.*

In this work, we deeply explore the application of multi-

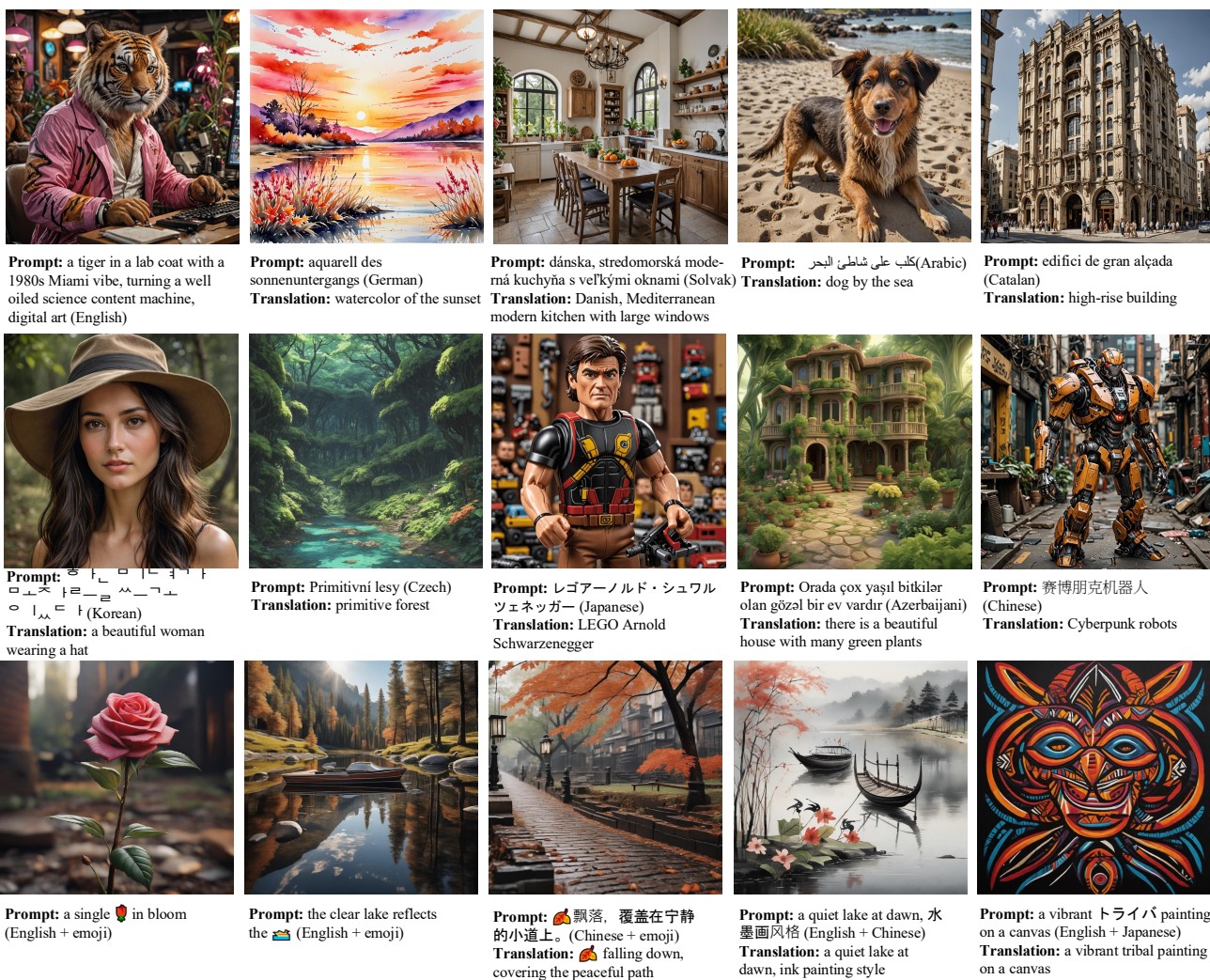

**Prompt:** a tiger in a lab coat with a 1980s Miami vibe, turning a well oiled science content machine, digital art (English)

**Prompt:** aquarell des sonnenuntergangs (German)
**Translation:** watercolor of the sunset

**Prompt:** dánska, stredomorská moderná kuchyňa s veľkými oknami (Solvak)
**Translation:** Danish, Mediterranean modern kitchen with large windows

**Prompt:** كلب على شاطئ البحر(Arabic)
**Translation:** dog by the sea

**Prompt:** edifici de gran alçada (Catalan)
**Translation:** high-rise building

**Prompt:** ᄒ ᅡ ᆫ ᄆ ᅵ ᄋ ᅧ ᆻ ᄀ ᅥ ᆫ ᄆ ᅩ ᄌ ᅡ ᄅ ᅳ ᆯ ᄊ ᅳ ᆫ ᄋ ᅵ ᄉ ᅡ (Korean)
**Translation:** a beautiful woman wearing a hat

**Prompt:** Primitivní lesy (Czech)
**Translation:** primitive forest

**Prompt:** レゴアーノルド・シュワルツェネッガー (Japanese)
**Translation:** LEGO Arnold Schwarzenegger

**Prompt:** Orada çox yaşıl bitkilər olan gözəl bir ev vardır (Azerbaijani)
**Translation:** there is a beautiful house with many green plants

**Prompt:** 赛博朋克机器人 (Chinese)
**Translation:** Cyberpunk robots

**Prompt:** a single 🌹 in bloom (English + emoji)

**Prompt:** the clear lake reflects the 🏞️ (English + emoji)

**Prompt:** 🍂飘落，覆盖在宁静的小道上。(Chinese + emoji)
**Translation:** 🍂 falling down, covering the peaceful path

**Prompt:** a quiet lake at dawn, 水墨画风格 (English + Chinese)
**Translation:** a quiet lake at dawn, ink painting style

**Prompt:** a vibrant トライバ painting on a canvas (English + Japanese)
**Translation:** a vibrant tribal painting on a canvas

*Figure 1.* **Images generated by MuLan with different backbones**, such as Dreamshaper 8, Dreamshaper XL Lightning, and Pixart-Alpha, using a variety of languages or mixed-language prompts.

lingual semantic alignment in image generation from the perspective of language and image-text alignment. We also reveal that text encoders trained on large-scale multilingual image-text datasets with noisy data demonstrate remarkable data efficiency in multilingual image generation. Based on this insight, we introduce *MuLan*, a lightweight *Multi-Lan*guage adapter that connects text encoders to diffusion models with low-cost training, facilitating native support for hundreds of languages. Specifically, we employ a plug-and-play language adapter with fewer than 20 million parameters to bridge a frozen multilingual text encoder with a frozen diffusion model, which exhibits outstanding zero-shot performance on multilingual T2I generation tasks.

Our model offers advantages in terms of not only multilingual performance, but also training cost and adaptability compared to previous works. In a nutshell, MuLan leverages

large-scale pre-trained multilingual text encoders, reducing the need for extensive multilingual datasets. As a result, the model can efficiently adapt to over 110 languages using only a small amount of English training data, achieving generation quality comparable to English (e.g., CLIP similarity scores: 39.57 for English, average 39.61 for other languages). Additionally, as shown in Figure 4, MuLan 's plug-and-play architecture seamlessly integrates with existing model architectures and frameworks, greatly enhancing its compatibility with various community-driven tools and models, such as LoRA (Hu et al., 2021), LCM (Luo et al., 2023), ControlNet (Zhang et al., 2023), and IP-Adapter (Ye et al., 2023b).

In summary, our contribution is three folds:

(1) We investigate different possible ways to equip monolingual text-to-image models with multilingual ability, among

which we demonstrate that using a text encoder properly trained from noisy web-scale data (such as InternVL (Chen et al., 2023b)) is of great data efficiency.

(2) We propose *MuLan*, a plug-and-play lightweight language adapter that can be combined with any community models and tools for native multilingual generation in 110+ languages. Our method requires only English image-text pairs and avoids problems in previous attempts, including preparing datasets in a restricted number of languages, heavy computing budget, and inflexibility when combined with various community models/tools.

(3) We demonstrate our method's effectiveness and efficiency through thorough quantitative and qualitative experiments. Notably, it requires only about 12 hours of training on 8 A100 GPUs and 17M English data to perform very well on 110+ languages, with very close CLIP similarity scores of 39.57 (English) v.s. 39.61 (average of other languages).

## 2. Related Work

**Multilingual Diffusion Models.** Recent advancements have seen the rise of diffusion-based models, which have significantly improved image generation quality and diversity. Popular models such as SD series (Rombach et al., 2022; Esser et al., 2024; Podell et al., 2023), DALL-E (Ramesh et al., 2022; 2021), Imagen (Saharia et al., 2022) and Glide (Nichol et al., 2022) demonstrate photorealistic generation capabilities, yet they are primarily trained on English data and thus struggle with multilingual image generation. Although diffusion models using CLIP text encoders can generalize somewhat to Romance languages (e.g., French), their performance significantly degrades for languages outside this family, particularly East Asian languages such as Chinese. This limitation arises from the models' reliance on English-centric text encoders, such as CLIP (Radford et al., 2021) and T5 (Raffel et al., 2020), and the predominantly English training data.

Recent efforts have explored multilingual image generation by incorporating multilingual text encoders and datasets to overcome the limitations of English-centric models. One approach involves building models entirely from scratch with non-English data. For instance, Hunyuan-DiT (Li et al., 2024b) and Kolors (Team, 2024) incorporate Chinese text encoders and extensive Chinese datasets to enhance their support for culturally specific concepts and improve generation quality in Chinese. Alternatively, some methods attempt to adapt existing models by replacing or fine-tuning the text encoder. Models such as Taiyi (Wu et al., 2024), PanGu (Lu et al., 2023), and AltDiffusion (Ye et al., 2023a) replace the text encoder in SD and then fine-tune it with multilingual data, thus reducing the overhead compared to training from scratch. GlueGen (Qin et al., 2023) and

PEA-Diffusion (Ma et al., 2023) focus on fully utilizing the current T2I model by employing lightweight networks to align the semantics of other languages with English using low-cost parallel data. However, this often involves complex distillation loss, making the training process less stable. In our work, multilingual image generation capabilities can be generalized simply by training with the commonly used SD loss on English image-text pairs.

**Multilingual Language Model.** Language models have evolved through various modeling approaches, reflecting a range of training objectives and capabilities. Early models, such as BERT (Devlin et al., 2019), BART (Lewis et al., 2019), focus on understanding sentence structure and contextual relationships by predicting masked tokens within a sentence. More recent large language models (LLMs), such as LLaMA series (Touvron et al., 2023; Dubey et al., 2024), T5 (Raffel et al., 2020) and GPT series (Brown et al., 2020; OpenAI, 2023), build on this foundation with improved context understanding and generation abilities, allowing them to handle a wide range of tasks with nuanced language comprehension and open-ended text generation. Building on these foundational text encoders, recent works, such as CLIP (Radford et al., 2021), ALIGN (Jia et al., 2021), and InternVL (Chen et al., 2023b), incorporate visual alignment through paired image-text data, creating joint representations that bridge language and vision for cross-modal tasks.

Previous works such as mBERT (Devlin et al., 2019) and LLaMA3 (Dubey et al., 2024) have shown strong multilingual capabilities by pre-training on large multilingual corpora, enabling these models to understand and generate text in a variety of languages. InternVL-LLaMA (Chen et al., 2023b) achieves powerful multilingual cross-modal capabilities by aligning a text encoder with a vision transformer (ViT) on the multilingual image-text dataset LAION(Schuhmann et al., 2022), enabling effective image-text contrastive learning. However, due to the scarcity of large-scale multilingual image-text datasets, some approaches (Chen et al., 2022; Carlsson et al., 2022) have resorted to using translated datasets and distillation learning to align multilingual text features. Despite these advances, an open challenge remains inefficiently leveraging multilingual text encoders for text-to-image (T2I) generation. Previous works, such as Taiyi (Wu et al., 2024) and AltDiffusion (Ye et al., 2023a), have adopted multilingual text encoders and fine-tuned them on multilingual image-text pairs for T2I tasks. In contrast, our method takes advantage of InternVL-LLaMA's multilingual capabilities, requiring only a small amount of English data and without the need to fine-tune the SD model weights, achieving state-of-the-art performance in multilingual T2I generation.

| Text Encoder | Architecture | Supervision Method | Aligned |
|---|---|---|---|
| BERT | encoder only | MLM | ✗ |
| T5 | encoder-decoder | MLM | ✗ |
| LLaMA | decoder only | NTP | ✗ |
| CLIP | encoder only | CL | ✓ |
| InternVL-LLaMA | decoder-only | NTP&CL | ✓ |

*Table 1.* **Common Text Encoders**. The common language model architectures and their supervision methods, "Aligned" indicates whether they have been aligned with images. In "Supervision Method", "MLM" is Masked Language Modeling, "NTP" is Next Token Prediction, and "CL" is Contrastive Learning.

# 3. Proposed Method

In this section, we first revisit the mainstream text-to-image generation framework, analyzing its underlying structures and presently available data resources. Based on these insights, we introduce MuLan, a cost-effective multilingual generation framework designed to improve cross-language adaptability and generation quality.

## 3.1. Revisiting Text-to-Image Generation

Given an input text prompt $x$ and ground-truth image $y$ from the training dataset $D$, a mainstream text-to-image (T2I) generation model $G(\cdot)$, which consists of a language model $L(\cdot)$ and a visual generator $V(\cdot)$, can be defined as follows:

$$\theta_l^*, \theta_v^* = \arg\min_{\theta_l, \theta_v} \mathbb{E}_{(x,y)\sim D}\left[\mathcal{L}(G(x;\ \theta_l, \theta_v), y)\right], \quad (1)$$

where $\theta_l$ and $\theta_v$ represent the parameters of the language model $L(\cdot)$ and the visual generator $V(\cdot)$, respectively, and $\mathcal{L}$ denotes the generation loss, $*$ represents the optimal solution for function optimization. It can be observed that the primary components related to multilingual processing are the training dataset $D$ and the language model $L(\cdot)$. Therefore, in the following sections, we focus on the two modules, exploring pathways for building an efficient text-to-image generation model.

**Language Model.** Existing T2I models typically employ pre-trained language models as text encoders. For instance, the SD series (Rombach et al., 2022; Podell et al., 2023; Esser et al., 2024) leverages CLIP (Radford et al., 2021) or T5 (Raffel et al., 2020). Recently, decoder-only LLMs like GPT (Brown et al., 2020; OpenAI, 2023) and LLaMA (Touvron et al., 2023; Dubey et al., 2024) have gained attention for their superior NLP performance, multilingual capabilities, and next-word prediction training. As summarized in Table 1, a diverse range of language models is available, yet *the optimal choice for multilingual T2I generation remains an open question*.

**Dataset.** As shown in Table 2, existing datasets—including

| name | num | type | lang-num | quality |
|---|---|---|---|---|
| LAION-5B | 5B | TI pairs | 100+ | Noisy |
| PixArt | 15M | TI pairs | 1 | High |
| JourneyDB | 4M | TI pairs | 1 | High |
| CCAligned | 100M | Text Parallel | 137 | Noisy |
| CCMatrix | 69B | Text Parallel | 80+ | Noisy |

*Table 2.* **Mainstream Dataset Types.** The existing data used for T2I training mainly includes two formats: Text-Image pairs (TI pairs) and Text Parallel data.

multilingual text corpora, multilingual text-image pairs, and high-quality text-to-image datasets—offer valuable resources for multilingual image generation. Large-scale datasets like Laion-400M/5B (Schuhmann et al., 2022) contain noisy, lower-quality data but provide multilingual text-image pairs, supporting diverse languages. Meanwhile, multilingual translation corpora (e.g., CCMatrix (Schwenk et al., 2020)) lack image-text pairs but offer cross-language alignments. High-quality text-to-image datasets (e.g., JourneyDB (Pan et al., 2023)) supply high-resolution images but mainly support English and a few mainstream languages. Thus, *leveraging low-cost, internet-based text-image datasets and multilingual translation corpora to enhance multilingual T2I model training remains a promising direction*.

## 3.2. MuLan: Toward Multilingual T2I Generation

**Overall Architecture.** To facilitate efficient and effective multilingual T2I generation, MuLan incorporates two key designs: (1) the multilingual semantic alignment through easily accessible large-scale data, and (2) a language adapter trained on a limited set of English T2I data. These designs enable MuLan to operate without the constraints of multilingual T2I data, allowing for more efficient training by leveraging existing models and datasets.

**Multilingual Semantic Alignment.** While many existing language models demonstrate robust multilingual capabilities, not all are well-suited for the multilingual T2I generation task needed for the Mulan framework. In this work, we emphasize the importance of maintaining a consistent vector space across languages for multilingual T2I generation. Specifically, given two text prompts, $x_1$ and $x_2$, that share the same meaning but are in different languages, and a text encoder $L(\cdot)$, the representations of these prompts, $L(x_1)$ and $L(x_2)$, should closely align. This alignment ensures that the conditional inputs to the image decoder remain consistent across languages, thereby preserving consistent image generation quality. Here, we mainly consider two alignment approaches: (1) Image-centered alignment; (2) Language-centered alignment.

(1) *Language-Centered Alignment.* A simple approach to multilingual semantic alignment is leveraging large-scale

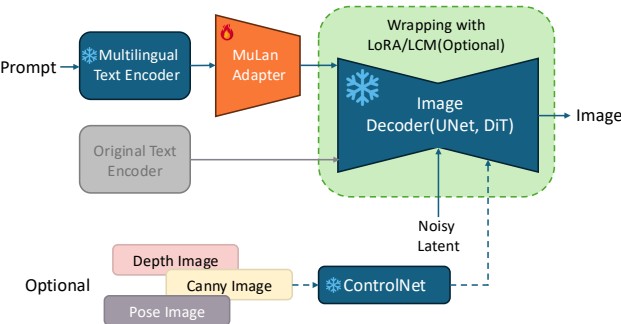

*Figure 2.* **Overview of of MuLan**. We use a language model as the Multilingual Text Encoder, which has undergone Multilingual Semantic Alignment stage. We only train the Language Adapter, while all other modules are frozen.

translation data to align other languages' vector spaces with the well-supported English vector space. By conducting distillation training with translation data alone, this can be achieved: we designate the SD's language encoder as the teacher encoder, indicated by $L_t(x)$ and any multilingual encoder $L_s(x, \theta_s)$ as the student encoder, aligning their features using MSE Loss. This alignment can be written as:

$$\theta_s^* = \arg \min_{\theta_s} \mathbb{E}_{(x,y) \sim D_{\mathrm{tr}}} [\mathrm{MSE}(L_s(x_1, \theta_s), L_t(x_2))] \quad (2)$$

(2) *Image-Centered Alignment.* In the image-centered alignment approach, CLIP maximizes the similarity between positive text-image pairs and minimizes it for negative pairs through contrastive learning. This training uses text-image pairs, and when the text includes multiple languages, the language encoder aligns different languages in the vector space naturally around the image. In this case, the objective function can be written as:

$$\theta_l^* = \arg \min_{\theta_l} \mathbb{E}_{(x,y) \sim D} [\mathrm{cosine}(L(x, \theta_l), E_I)], \quad (3)$$

Here, $\theta_l$ refers to the parameters of the language model $L(\cdot)$, $E_I$ refers to the image feature. This method, however, is resource-intensive and requires large multilingual text-image pairs, which are challenging to obtain and require substantial storage. Using pre-trained models can help reduce these costs.

**Language Adapter.** After getting the aligned language model, to achieve cost-effective multilingual text-to-image generation, we propose MuLan. This model incorporates a lightweight language adapter $L'$ that bridges a multilingual-aligned language model with a visual generator, enabling generalization to multiple languages after training on a small amount of English text-to-image generation data $D_{\mathrm{en}}$. So the Eqn 1 could be rewritten as:

$$\theta_{l'}^* = \arg \min_{\theta_{l'}} \mathbb{E}_{(x,y) \sim D_{\mathrm{en}}} [\mathcal{L}(G(x; \theta_{l'}), y)]. \quad (4)$$

After aligning different languages using Eqn 2 or Eqn 3, we can achieve multilingual T2I generation by training only this adapter. The key to low-cost implementation lies in freezing the language model and visual generator while training only the language adapter $\theta_{l'}$ on small-scale English data $D_{\mathrm{en}}$.

These adapters are used to re-project the high-dimensional representations of text prompts from different languages into a unified space. We adopt different adapter designs for different diffusion models. In detail, we can achieve good results using a simple MLP architecture for Pixart-$\alpha$ (Chen et al., 2023a). However, we find MLP could not properly deal with SD models (Rombach et al., 2022; Podell et al., 2023). Instead, we choose to use one layer encoder-decoder transformer with a set of learnable queries for extracting embeddings from InternVL outputs. For SDXL (Podell et al., 2023), we use two transformers to project embeddings for two text encoders that SDXL adopts, and one attention pooling layer for extracting pool embeddings.

**Bridge Together.** As shown in Figure 2, after the Multilingual text encoder is trained with Multilingual Semantic Alignment, the Language Adapter can be trained with minimal English training data at a low cost while enabling multilingual image generation capabilities for broad text-to-image diffusion models. The Multilingual Semantic Alignment stage can be undergone with different strategies and base models, which could result in different performance on subsequent image-generation fine-tuning. More details are elucidated in Section 4.

## 4. Experiments

### 4.1. Experimental setup

**Datasets.** We primarily use a subset of LAION-EN (Schuhmann et al., 2022) with all samples that have aesthetic scores larger than 5.8 for training base models and PixArt (Chen et al., 2023a) dataset for aesthetic models.

**Implementation details.** We use InternVL-LLaMA (Chen et al., 2023b) as our text encoder, which has undergone Image-Centered Alignment on a multilingual image-text pair dataset. Then, we trained the MuLan Adapter for different image decoders using the architecture described in Section 3.

All MuLan adapters are trained with AdamW (Loshchilov & Hutter, 2017) optimizer and 128 batch size. We use constant learning rate 1e-5 for SD 1.5/2.1 (Rombach et al., 2022), 1e-6 for SDXL (Podell et al., 2023), and 2e-5 for Pixart-$\alpha$ (Chen et al., 2023a). For SD 1.5, we train the adapter for 50k steps at the resolution of $512 \times 512$, and for SD 2.1 we adjust the resolution to $768 \times 768$. For SDXL, we first train the adapter for 100k steps at the resolution of $512 \times 512$ and finetune it for another 1k steps at the resolution of

1024×1024. For Pixart-α, we train the adapter for 118k steps at the resolution of 512×512. We randomly drop text conditions at the rate of 10% and use min-SNR (Hang et al., 2023) to accelerate training.

The training process was conducted on 8 NVIDIA A100-80G GPUs. For SD 1.5 and SD 2.1, it took 12 hours, and for SDXL and Pixart-α, it took two days.

**Evaluation Metrics.** We use XM3600 (Thapliyal et al., 2022) to assess the model's capabilities across 12 mainstream languages (denoted as XM12 hereafter). To evaluate the model's generalization to additional languages, we tested multilingual versions of the COCO2014 (Lin et al., 2015) validation set. We translated the prompts into 85 languages using Google Translate for these datasets. The model's performance was compared with an ad-hoc translation-based SD 1.5 model and other multilingual T2I models (Yan et al., 2024). Regarding evaluation metrics, we employed FID and CLIP Score (SIM) calculated by InternVL-LLaMA (Chen et al., 2023b).

## 4.2. Quantitive Results

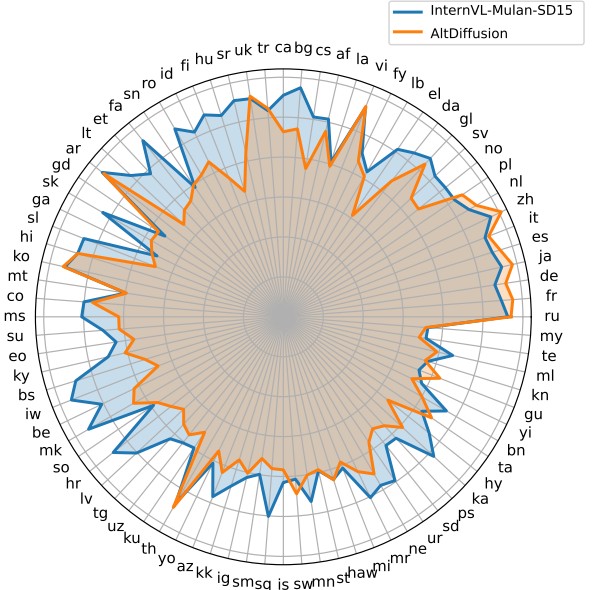

*Figure 3.* **Comparison of CLIP Score of InternVL-MuLan-SD15 and AltDiffusion across hundreds of languages on COCO2014 val.** Our model achieved performance comparable to AltDiffusion in mainstream languages while substantially surpassing AltDiffusion in less common languages.

**Multilingual T2I Comparison**. We integrated InternVL-LLaMA (Chen et al., 2023b) into the adapter's model, which we call *InternVL-MuLan*. Specifically, for each image decoder, we train a separate adapter model, which we refer to as InternVL-MuLan-SD15, InternVL-MuLan-SD21, InternVL-MuLan-SDXL and InternVL-MuLan-PixArt.

As shown in Table 3, we compared our model with other multilingual image generation models. On the XM12 dataset, InternVL-MuLan-PixArt outperformed AltDiffusion (Ye et al., 2023a) and GlueGen (Qin et al., 2023) in CLIP score, highlighting its strong multilingual understanding. However, InternVL-MuLan-PixArt fell behind AltDiffusion in FID. To further evaluate image quality, we used Laion's aesthetic predictor. InternVL-MuLan-SD15 and InternVL-MuLan-PixArt achieved average aesthetic scores of 6.31 and 6.67, respectively, surpassing AltDiffusion's score of 6.05. This highlights the superior quality and visual appeal of images generated by our model.

To further evaluate our model's capability in the broader range of minority languages, we evaluated our model on the COCO2014 (Lin et al., 2015) validation set (85 languages) and computed CLIP Score, as shown in Figure 3. Our model achieved performance comparable to AltDiffusion in mainstream languages while substantially surpassing AltDiffusion in less common languages.

Our model achieves top performance across languages, reaching SOTA levels in mainstream languages and excelling in low-resource ones, all with minimal training costs. InternVL-MuLan-SD15 requires just 0.5×8 GPU-days, and InternVL-MuLan-PixArt only 2×8 GPU-days. In comparison, GlueGen costs 5 GPU-days yet performs worse than the low-cost InternVL-MuLan-SD15, while AltDiffusion demands at least 19×64 GPU-days. This highlights the efficiency and cost-effectiveness of our approach.

**Comparison of MuLan and Translate API** We compared MuLan with the approach of using Google Translate and NLLB (NLLB Team et al., 2022) to convert non-English languages into English for input into image generation models (Sun et al., 2024). Non-English languages in XM12 were translated into English, as input of SD15 (Rombach et al., 2022) and PixArt-α (Chen et al., 2023a), with results compared to direct input into MuLan. As shown in Table 3, MuLan outperforms the approach that uses the open-source model NLLB as a translation tool and achieves performance comparable to using Google Translate. It demonstrates even better results in low-resource languages where translation tools often struggle to provide accurate translations.

**Comparison of MuLan and "as-is" Baseline** We evaluated Stable Diffusion v1.5 and PixArt-α by directly inputting prompts in 11 languages from the XM12 dataset, using InternVL-LLaMA to compute CLIP scores, shown in Table 5. These "as-is" results, where the prompts in different languages were fed into the T2I models without translation, were then compared with our MuLan-adapted models. We found that MuLan adapters consistently improved performance across all languages. Notably, even for languages close to English—such as French, Spanish, and German—our method still achieved clear gains over the

| | | avg. | en | fr | es | it | zh | ja | hi | de | ko | ru | ar | pl |
|---|---|---|---|---|---|---|---|---|---|---|---|---|---|---|
| **GlueGen\*** | SIM | 35.6 | 39.6 | 35.9 | 35.5 | 35.0 | 35.3 | 36.3 | ✗ | ✗ | ✗ | ✗ | ✗ | ✗ |
| | FID | 17.4 | 21.4 | 17.1 | 18.5 | 17.6 | 17.6 | 16.1 | ✗ | ✗ | ✗ | ✗ | ✗ | ✗ |
| **AltDiffusion** | SIM | 38.8 | 40.0 | 40.0 | 39.4 | 39.7 | 39.0 | 39.4 | 34.1 | 39.0 | 38.5 | 38.6 | 38.3 | 39.2 |
| | FID | **9.1** | 8.1 | 8.2 | 9.4 | 10.4 | 9.4 | 9.4 | 10.2 | 8.2 | 8.9 | 9.7 | 8.9 | 9.1 |
| **SD15(NLLB)** | SIM | 36.5 | 39.6 | 37.6 | 37.4 | 37.2 | 36.2 | 36.2 | 32.7 | 37.7 | 35.9 | 36.8 | 33.9 | 36.3 |
| | FID | 25.0 | 21.4 | 24.9 | 25.0 | 25.0 | 25.5 | 24.9 | 28.2 | 24.5 | 25.4 | 24.6 | 26.6 | 24.4 |
| **SD15(Google)** | SIM | 36.9 | 39.6 | 38.2 | 38.0 | 37.8 | 36.6 | 36.7 | 33.1 | 38.4 | 36.4 | 37.4 | 34.4 | 36.7 |
| | FID | 21.6 | 21.4 | 21.6 | 23.9 | 22.4 | 19.1 | 20.5 | 19.6 | 22.1 | 21.7 | 21.6 | 22.9 | 21.8 |
| **PixArt(NLLB)** | SIM | 38.3 | 39.0 | 39.7 | 39.4 | 39.2 | 38.1 | 38.6 | 34.4 | 40.3 | 37.7 | 39.3 | 35.6 | 38.7 |
| | FID | 13.2 | 12.6 | 13.2 | 14.6 | 12.5 | 13.0 | 13.1 | 12.4 | 12.9 | 12.4 | 13.0 | 14.3 | 14.7 |
| **PixArt(Google)** | SIM | **39.7** | 39.0 | 41.2 | 40.6 | 41.0 | 39.9 | 40.8 | 34.7 | 41.5 | 39.0 | 40.5 | 39.0 | 39.9 |
| | FID | 13.7 | 12.6 | 13.7 | 15.5 | 12.9 | 13.8 | 13.9 | 12.3 | 12.9 | 13.2 | 12.9 | 15.1 | 15.4 |
| **Mulan-SD15** | SIM | 37.7 | 38.6 | 38.0 | 37.7 | 37.8 | 37.7 | 38.0 | 35.6 | 38.0 | 37.6 | 37.9 | 38.2 | 37.1 |
| | FID | 14.4 | 13.9 | 14.7 | 16.5 | 13.5 | 17.0 | 13.7 | 13.0 | 13.8 | 12.8 | 13.8 | 14.0 | 16.5 |
| **Mulan-PixArt** | SIM | 39.5 | 39.6 | 40.2 | 39.6 | 40.0 | 39.3 | 40.5 | 37.2 | 40.5 | 39.3 | 39.6 | 39.1 | 39.1 |
| | FID | 11.8 | 12.8 | 12.2 | 13.6 | 11.7 | 10.6 | 10.7 | 13.4 | 12.4 | 8.4 | 10.5 | 11.9 | 13.2 |

*Table 3.* **Comparison of Multilingual Generation Capabilities on XM12.** GlueGen and AltDiffusion are other multilingual T2I methods, where the inputs for SD15 and PixArt-$\alpha$ are translated using Google Translate and NLLB. GlueGen is equivalent to the original SD15 when accepting English prompts. Its performance in English was not included in the average calculation.

| lang Id | Model | SIM |
|---|---|---|
| zh | Taiyi-SDXL-3.5B | 35.44 |
| | Taiyi-SD-1B-Chinese | 36.84 |
| | **InternVL-Mulan-SD15** | **37.84** |
| ja | JapaneseSDXL | 38.00 |
| | **InternVL-Mulan-SD15** | **38.01** |

*Table 4.* **Comparison of CLIP Score on Chinese and Japanese with specialized models on XM3600.** InternVL-MuLan-SD15 surpasses specialized Chinese and Japanese models.

as-is baseline, demonstrating its broad effectiveness.

**Alternative CLIP for Image-Text Similarity** Image-text similarity is typically computed using the industry-standard CLIP-ViT models (Radford et al., 2021). However, these models are primarily trained on English data, making their similarity scores unreliable for non-English inputs. In this work, we employ InternVL-LLaMA, a multilingual model, to compute the image-text similarity scores. To further ensure the objectivity of the similarity evaluation, we additionally use the multilingual CLIP-ViT model released by Laion (LAION, 2023). The results shown in Table 5 demonstrate that our model remains robust, and the observed trends are consistent with those reported in Table 3.

**Fine-grained Text-Image Alignment Evaluation** While CLIP Score is useful for evaluating the presence of main subjects, it is limited in capturing object-level details and spatial relations—especially for compositional prompts in XM12. To address this, we adopt VQAScore (Lin et al., 2024) with GPT-4o as the evaluator, enabling reliable multilingual assessment. As shown in Table 5, our model achieves strong fine-grained alignment, outperforming GlueGen and Alt-Diffusion, and matching or surpassing translation-based baselines across most languages.

**Comparison of MuLan and Specialized Models** We compared the multilingual MuLan adapter with two language-specific models: Taiyi (Wu et al., 2024), trained on millions of Chinese image-text pairs, and the Japanese SDXL model (Shing & Akiba), fine-tuned on high-quality Japanese image-text pairs. Our model performs better on both languages, demonstrating its competitiveness with specialized models.

**Comparison of Multilingual Semantic Alignment Methods** In this study, we trained the MuLan Adapter on multilingual language models with and without Multilingual Semantic Alignment, using different alignment methods, and evaluated their performance on five non-English languages on the XM12 dataset. The results are shown in Table 6. We first selected the multilingual-supporting LLama (Touvron et al., 2023), mT5-xl (Xue et al., 2021) and XLM-R Large (Conneau, 2019) models as text encoders and trained MuLan without the Multilingual Semantic Alignment stage. Next, for Image-Centered Alignment, we choose InternVL-LLaMA (Chen et al., 2023b) and Mul-OpenCLIP (Conneau, 2019) as text encoders, both of which have be trained on multilingual image-text pairs such as Laion-5B (Schuhmann et al., 2022). For Language-Centered Alignment, we selected MultiLang-CLIP (Carlsson et al., 2022) and AltClip-m18 (Chen et al., 2022), models trained on parallel datasets like CCMatrix (Schwenk et al., 2020), We also trained an XLM-R Large on CCMatrix for comparison. Among them, Mul-OpenCLIP, MultiLang-CLIP, and AltClip-m18 all use the XLM-Roberta-Large architecture, but with different training methods or datasets. All image decoders used are based on SD 1.5 (Rombach et al., 2022).

The results are shown in Table 6. We found that multilingual text encoders trained solely on multilingual corpora, such

| | Metric | avg. | en | fr | es | it | zh | ja | hi | de | ko | ru | ar | pl |
|---|---|---|---|---|---|---|---|---|---|---|---|---|---|---|
| **SD15(as-is)** | SIM | 29.1 | 39.6 | 34.2 | 34.2 | 31.8 | 26.6 | 26.9 | 26.8 | 31.9 | 24.6 | 23.0 | 24.5 | 24.8 |
| **PixArt(as-is)** | SIM | 29.8 | 39.0 | 38.2 | 36.9 | 36.0 | 24.2 | 22.1 | 24.0 | 36.8 | 22.4 | 28.9 | 22.2 | 27.0 |
| **Mulan-SD15** | SIM | **37.7** | 38.6 | 38.0 | 37.7 | 37.8 | 37.7 | 38.0 | 35.6 | 38.0 | 37.6 | 37.9 | 38.2 | 37.1 |
| **Mulan-PixArt** | SIM | **39.5** | 39.6 | 40.2 | 39.6 | 40.0 | 39.3 | 40.5 | 37.2 | 40.5 | 39.3 | 39.6 | 39.1 | 39.1 |
| | | | | | | | | | | | | | | |
| **GlueGen** | SIM(Laion) | 21.1 | 22.3 | 21.3 | 20.6 | 19.6 | 21.5 | 21.0 | ✗ | ✗ | ✗ | ✗ | ✗ | ✗ |
| **AltDiffusion** | SIM(Laion) | 23.1 | 24.2 | 23.6 | 23.2 | 23.1 | 24.3 | 23.0 | 21.1 | 24.3 | 22.5 | 22.3 | 22.7 | 23.2 |
| **SD15(Google)** | SIM(Laion) | 22.3 | 22.3 | 23.8 | 23.4 | 22.8 | 22.6 | 21.3 | 20.8 | 24.5 | 21.7 | 22.6 | 19.5 | 21.9 |
| **PixArt(Google)** | SIM(Laion) | **24.3** | 24.1 | 24.2 | 23.8 | 24.5 | 26.4 | 25.1 | 21.6 | 26.5 | 23.1 | 24.6 | 23.3 | 23.9 |
| **Mulan-SD15** | SIM(Laion) | 23.0 | 21.8 | 23.6 | 23.2 | 22.7 | 23.6 | 24.2 | 22.7 | 24.5 | 21.8 | 23.0 | 22.3 | 22.9 |
| **Mulan-PixArt** | SIM(Laion) | 24.2 | 24.4 | 23.9 | 23.4 | 24.2 | 25.7 | 24.8 | 23.2 | 25.8 | 23.4 | 24.6 | 23.1 | 23.4 |
| | | | | | | | | | | | | | | |
| **GlueGen** | VQAScore | 0.53 | 0.81 | 0.51 | 0.52 | 0.45 | 0.41 | 0.50 | ✗ | ✗ | ✗ | ✗ | ✗ | ✗ |
| **AltDiffusion** | VQAScore | 0.58 | 0.71 | 0.60 | 0.60 | 0.59 | 0.50 | 0.53 | 0.62 | 0.49 | 0.56 | 0.51 | 0.64 | 0.62 |
| **SD15(Google)** | VQAScore | 0.61 | 0.81 | 0.66 | 0.71 | 0.61 | 0.57 | 0.50 | 0.67 | 0.50 | 0.70 | 0.68 | 0.41 | 0.52 |
| **PixArt(Google)** | VQAScore | **0.75** | 0.85 | 0.78 | 0.80 | 0.82 | 0.71 | 0.71 | 0.82 | 0.71 | 0.76 | 0.76 | 0.59 | 0.69 |
| **Mulan-SD15** | VQAScore | 0.64 | 0.80 | 0.74 | 0.76 | 0.68 | 0.60 | 0.52 | 0.53 | 0.57 | 0.65 | 0.75 | 0.45 | 0.57 |
| **Mulan-PixArt** | VQAScore | 0.74 | 0.88 | 0.81 | 0.83 | 0.81 | 0.71 | 0.71 | 0.61 | 0.73 | 0.74 | 0.79 | 0.59 | 0.72 |

*Table 5.* **Multidimensional Comparison of Multilingual Image Generation on XM12.** *Part 1:* Comparison between English-only T2I models using as-is prompts and MuLan; *Part 2:* Using the multilingual-capable Laion's CLIP-ViT (LAION, 2023) for image-text similarity; *Part 3:* Evaluating fine-grained text-image alignment using VQAScore (Lin et al., 2024).

| Model | Align Method | avg SIM |
|---|---|---|
| LLaMA2-7B | None | ✗ |
| mT5-xl | None | ✗ |
| XLM-R Large | None | ✗ |
| MultiLang-CLIP | LC | 33.2 |
| AltClip-m18 | LC | 33.3 |
| XLM-R Large* | LC | 34.7 |
| Mul-OpenCLIP | IC | 36.1 |
| InternVL-LLaMA | IC | **37.8** |

*Table 6.* **CLIP Score on five non-English languages of Different Text Encoders trained with MuLan Adapter.** Mul-OpenCLIP stands for CLIP-ViT-H-14-frozen-xlm-roberta-large-laion5B-s13B-b90k (LAION, 2023). "LC" and "IC" represent Language-Centered Alignment and Image-Centered Alignment, respectively; XLM-R Large* is trained by the authors using CC-Matrix.

as LLaMA, mT5-xl, and XLM-R Large, fail to understand non-English prompts entirely. This is because these models have not undergone multilingual alignment, so their multilingual embeddings are not aligned. After connecting these language models with the MuLan adapter and training with English text-image pairs, only the English embeddings are aligned with the visual feature space, preventing understanding of non-English prompts. However, with either of the two alignment methods, the multilingual embeddings are first mapped to a unified feature space, which the MuLan adapter then aligns with the image space using only English image-text pairs, enabling multilingual image generation.

The results also suggest that Image-Centered Alignment outperforms Language-Centered Alignment by a significant margin. We speculate that this is because, in Image-Centered Alignment, language alignment is achieved

through the image, which acts as a bridge between the language embeddings and the visual space. As a result, the semantic space of different languages is already aligned with the image prior to training with MuLan, allowing MuLan to further refine this alignment. In contrast, Language-Centered Alignment relies entirely on MuLan to establish the connection between language embeddings and the image feature space, making the MuLan adapter training more challenging and resulting in lower performance.

In summary, MuLan innovatively uses an image-centered aligned language model for multilingual image generation. Previous methods like GlueGen (Qin et al., 2023) and AltDiffusion (Ye et al., 2023a), which rely on language-centered alignment, face inherent limitations due to their alignment method. GlueGen aligns non-English embeddings with the CLIP text encoder used in SD, with SD kept frozen, but as shown in Table 3, its performance on non-English languages is significantly worse than that on English. AltDiffusion, by contrast, retrains both text encoder and UNet on billions of multilingual pairs, incurring high training costs. In comparison, our approach achieves strong, balanced multilingual generation with minimal cost, requiring only a small adapter while keeping the SD frozen. In practice, the use of a unified, image-centered alignment pre-trained model allows easy adaptation to various SD models in the community with minimal overhead.

### 4.3. Qualitative Results

Our model can generate high-quality images across multiple languages. It also supports multilingual mixed input and can recognize certain emojis. In addition to simple T2I tasks, there are numerous downstream applications of SD

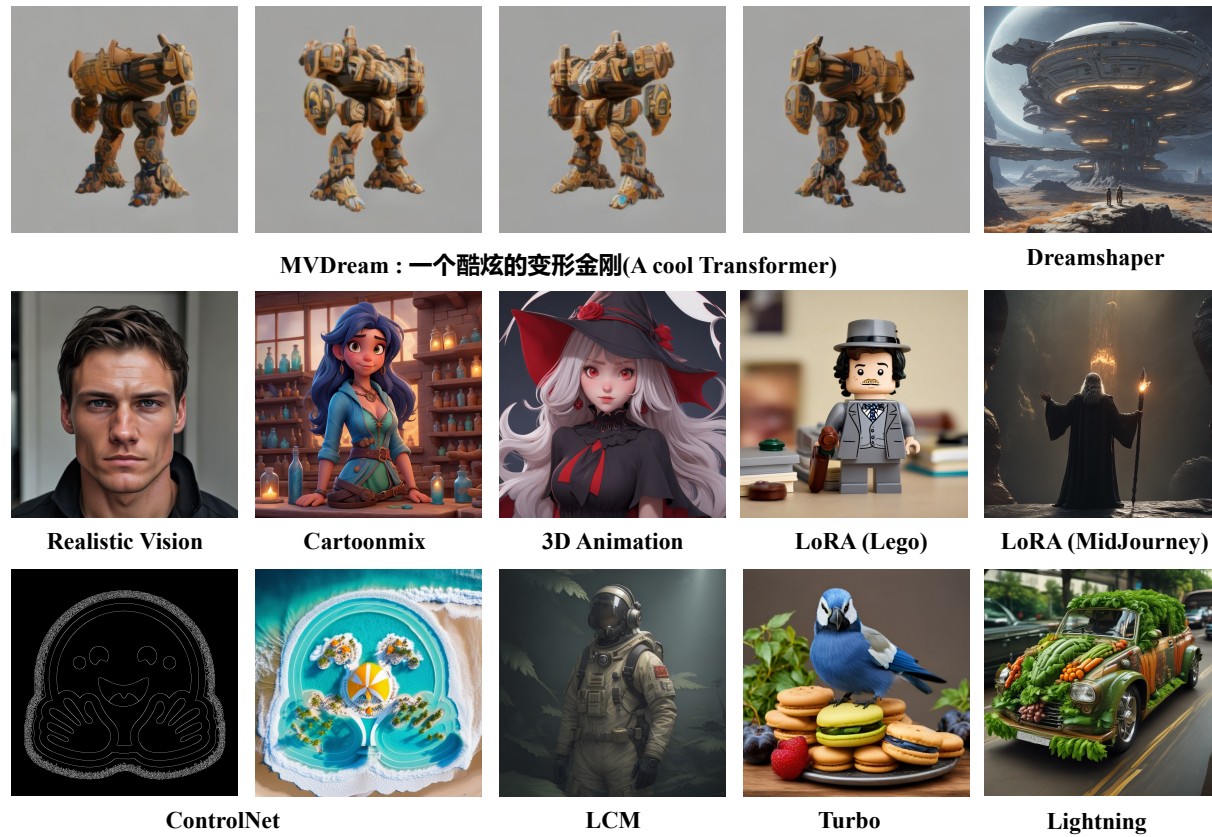

**MVDream : 一个酷炫的变形金刚(A cool Transformer)**

*Figure 4.* **Examples of MuLan integrates with community tools.** MuLan seamlessly integrates with MVDream (Shi et al., 2023) for 3D model generation and is fully compatible with community tools like LoRA (Hu et al., 2021), ControlNet (Zhang et al., 2023), and LCM (Luo et al., 2023).

within the community, including LoRA (Hu et al., 2021), LCM (Luo et al., 2023), and ControlNet (Zhang et al., 2023). These tools played crucial roles in enhancing model adaptability, control over outputs, and finetuning for specific tasks. Our model does not require finetuning on SD, allowing it to seamlessly integrate and be compatible with these community-developed SD applications in a plug-and-play manner. We show some examples in Figure 4.

## 5. Conclusion

We introduce language adapter **MuLan** that could equip image/video/3D diffusion models with multilingual generation abilities. MuLan shows strong zero-shot capabilities for up to 110 different languages, even if the adapter is solely trained on English data. MuLan also can be trained with a frozen text encoder and diffusion denoising model, which makes it applicable for many downstream models, such as LoRA (Hu et al., 2021), ControlNet (Zhang et al., 2023), LCM (Luo et al., 2023), and *etc.*, without any additional finetuning. MuLan is currently trained with paired data, and it could inevitably bring in bias and cause a distribution shift

of original models. A promising extension would be to alleviate the need for paired data and make original capabilities intact. Furthermore, MuLan currently focuses on improving multilingual generation capabilities, but it would be interesting to extend it to improving prompt understanding and following under a multilingual context.

## Impact Statement

This paper presents work whose goal is to advance the field of Machine Learning. There are many potential societal consequences of our work, none which we feel must be specifically highlighted here.

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

# A. More Experiments

In this section, we provide supplementary analyses to further evaluate and understand the performance and efficiency of our proposed model. These studies delve into key aspects such as training efficiency, semantic alignment methods, and the model's generalization across multiple languages. By examining the impact of various design choices and alignment strategies, we aim to shed light on the model's multilingual capabilities and resource efficiency.

## A.1. Training Efficiency

Our model leverages the multilingual capabilities of the InternVL-LLaMA text encoder, enabling effective training with minimal cost. In this section, we examine the changes in the model's performance when data volume and training iterations are reduced.

| $\lambda$ | # of samples | Avg SIM |
|---|---|---|
| 1 | 17M | 35.80 |
| 1/2 | 8.5M | 35.46 |
| 1/4 | 4.25M | 35.40 |
| 1/8 | 2.1M | 35.23 |
| 1/16 | 1M | 35.17 |
| 1/64 | 250k | 35.13 |
| 1/256 | 63k | 35.08 |
| 1/1024 | 16k | 33.49 |

*Table 7.* **Impact of dataset size on multilingual performance of the model.** We evaluated the model's performance using different proportions of the original dataset, ranging from full size ($\lambda = 1$) to $1/1024$, and measured the average CLIP Score across 7 languages. The results show that the model maintains relatively strong performance even with significantly reduced data sizes, demonstrating its efficiency and robustness under resource-constrained settings.

**Setting.** We choose Stable Diffusion 1.5 (Rombach et al., 2022) as the backbone model, using the AdamW optimizer for training. The model was trained to convergence with a learning rate of 2e-5 and a batch size of 128, without employing any training tricks. Additionally, the dataset size was reduced, and we set multiple data proportion levels ranging from 0.5 to 0.001 to evaluate the impact of data size on the model's performance.

**Results.** As shown in Table 7, we found that the model's multilingual performance decreased as the data volume was reduced; however, it still maintained a relatively strong performance until we reduced the data to $1/1024$ of the default data volume. This level of data volume and training cost is highly developer-friendly, requiring only 48 GPU hours to achieve decent multilingual text-to-image (T2I) capabilities for the model. Compared to existing multilingual T2I models, such as AltDiffusion (Ye et al., 2023a), our approach

requires significantly less data and computational cost at every stage.

## A.2. Impact of Semantic Alignment on Multilingual Features

In this section, we conduct a preliminary analysis of the multilingual output features produced by various text encoders to reveal the effects of the two image alignment training methods introduced in Section 3.2. The text encoders we selected for this analysis include XLM-R Large (Conneau, 2019), CCMatrix trained XLM-R Large* in Table 6, Mul-OpenCLIP (LAION-5B (Schuhmann et al., 2022) pretrained XLM-RoBERTa-Large (Conneau, 2019)), LLaMA2-7B (Touvron et al., 2023), and InternVL-LLaMA (Chen et al., 2023b).

We employ t-SNE (van der Maaten & Hinton, 2008), a nonlinear dimensionality reduction technique, to visualize the aggregation effects of multilingual features produced by these text encoders. t-SNE is particularly suited for preserving the local structure of high-dimensional data in a low-dimensional space. Ideally, for text encoders trained with multilingual alignment methods, the features corresponding to semantically equivalent prompts in different languages should exhibit aggregation effects when projected into a lower-dimensional space.

For this analysis, we randomly sampled 20 captions from the COCO2014 (Lin et al., 2015) validation set and translated them into 8 languages using machine translation, resulting in a total of 160 textual inputs. These inputs were then encoded using the selected text encoders to obtain their corresponding pooled feature representations.

The results are shown in Figure 5. As shown in Figures (a) and (d), the sample points in the t-SNE (van der Maaten & Hinton, 2008) plots exhibit a scattered distribution for text encoders that have not undergone image alignment training. This indicates that although XLM-RoBERTa (Conneau, 2019) and LLaMA2 (Touvron et al., 2023) possess multilingual representation capabilities, their output features for synonymous multilingual prompts are not closely located in the Euclidean space. In contrast, as shown in Figures (b)(c)(e), the sample points exhibit a clustered distribution after alignment training with images, with each cluster corresponding to the 20 different language versions of a single prompt. Through alignment training with images, the model minimizes the semantic discrepancies between multilingual representations, ensuring that the embeddings of synonymous prompts converge in the vector space.

## A.3. Detailed Results on COCO2014 validation set

In Section 4.2, we evaluated InternVL-Mulan-SD15 on the COCO2014 (Lin et al., 2015) validation set (85 languages)

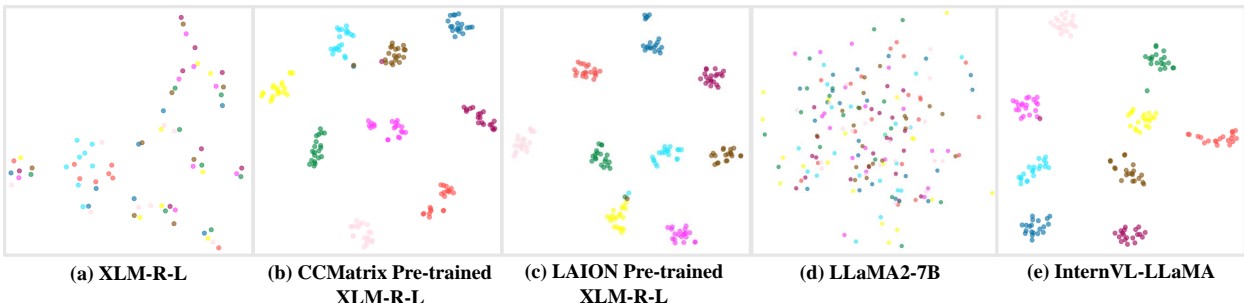

|  |  |  |  |  |
| --- | --- | --- | --- | --- |
| **(a) XLM-R-L** | **(b) CCMatrix Pre-trained XLM-R-L** | **(c) LAION Pre-trained XLM-R-L** | **(d) LLaMA2-7B** | **(e) InternVL-LLaMA** |

*Figure 5.* **t-SNE analysis on embeddings of 9 prompts in 20 languages produced by 5 text encoders**. (a) XLM-RoBERTa-Large (Conneau, 2019) (b) CCMatrix pre-trained XLM-RoBERTa-Large (Conneau, 2019) in Table 6 (c) Laion-5B (Schuhmann et al., 2022) pre-trained XLM-RoBERTa-Large (Conneau, 2019) (d) LLaMA2-7B (Touvron et al., 2023) (e) InternVL-LLaMA (Chen et al., 2023b). Points of the same color represent embeddings corresponding to the same prompt translated into different languages.

| Id | SD15 | SD21 | SDXL | AD | Id | SD15 | SD21 | SDXL | AD | Id | SD15 | SD21 | SDXL | AD |
| --- | --- | --- | --- | --- | --- | --- | --- | --- | --- | --- | --- | --- | --- | --- |
| af | 35.39 | 34.54 | 34.68 | 33.83 | id | 35.67 | 35.55 | 35.49 | 31.53 | ps | 30.74 | 30.87 | 31.41 | 28.61 |
| ar | 39.00 | 38.20 | 38.13 | 38.77 | ig | 30.65 | 30.80 | 32.10 | 30.09 | ro | 37.17 | 36.88 | 36.93 | 30.51 |
| az | 34.19 | 33.67 | 33.66 | 30.95 | is | 30.68 | 30.96 | 31.97 | 29.17 | ru | 38.11 | 37.86 | 37.63 | 38.51 |
| be | 35.24 | 35.23 | 35.20 | 30.74 | it | 37.13 | 36.62 | 36.56 | 37.62 | sd | 30.61 | 30.20 | 30.26 | 27.93 |
| bg | 38.76 | 38.40 | 38.38 | 33.60 | iw | 38.52 | 38.07 | 38.17 | 26.84 | sk | 36.12 | 36.12 | 36.02 | 29.09 |
| bn | 33.58 | 32.56 | 33.61 | 30.06 | ja | 38.05 | 38.14 | 37.82 | 39.43 | sl | 36.83 | 36.28 | 36.22 | 27.21 |
| bs | 37.20 | 36.89 | 36.86 | 28.10 | ka | 35.70 | 35.53 | 35.80 | 30.38 | sm | 29.94 | 30.24 | 32.44 | 27.94 |
| ca | 37.74 | 37.43 | 37.22 | 33.17 | kk | 32.18 | 31.70 | 31.44 | 28.75 | sn | 29.61 | 29.71 | 30.74 | 29.97 |
| co | 35.03 | 34.50 | 34.39 | 33.95 | km | 25.82 | 26.71 | 28.43 | 24.81 | so | 29.67 | 29.05 | 31.38 | 29.05 |
| cs | 35.30 | 34.85 | 34.71 | 28.78 | kn | 28.94 | 27.42 | 29.65 | 28.59 | sq | 35.08 | 34.41 | 34.77 | 29.06 |
| da | 36.25 | 35.99 | 35.77 | 31.78 | ko | 38.06 | 37.21 | 37.09 | 38.27 | sr | 37.77 | 37.90 | 37.91 | 31.09 |
| de | 36.79 | 36.98 | 36.87 | 38.16 | ku | 29.86 | 29.23 | 30.41 | 27.50 | st | 29.76 | 30.00 | 32.24 | 29.63 |
| el | 35.39 | 35.49 | 35.24 | 25.36 | ky | 32.49 | 32.08 | 32.37 | 30.33 | su | 32.65 | 32.00 | 32.91 | 30.62 |
| eo | 31.17 | 30.81 | 30.99 | 28.92 | la | 30.25 | 30.32 | 30.39 | 29.71 | sv | 35.91 | 36.22 | 36.07 | 31.83 |
| es | 37.38 | 37.41 | 37.37 | 38.41 | lb | 30.96 | 30.59 | 30.93 | 30.28 | sw | 30.39 | 30.30 | 30.77 | 32.23 |
| et | 34.43 | 34.85 | 34.76 | 28.22 | lt | 36.02 | 35.94 | 35.77 | 27.00 | ta | 31.00 | 29.69 | 32.20 | 32.47 |
| fa | 38.22 | 37.99 | 37.92 | 28.58 | lv | 35.08 | 34.91 | 34.56 | 27.03 | te | 28.07 | 27.22 | 29.82 | 27.21 |
| fi | 37.16 | 37.16 | 36.74 | 28.71 | mi | 30.34 | 30.60 | 32.75 | 29.54 | tg | 30.84 | 30.43 | 31.29 | 28.34 |
| fr | 37.38 | 37.57 | 37.41 | 38.81 | mk | 38.17 | 38.29 | 38.41 | 31.66 | th | 36.95 | 36.83 | 36.76 | 37.56 |
| fy | 32.73 | 31.97 | 31.90 | 31.64 | ml | 31.75 | 29.67 | 32.78 | 29.72 | tr | 35.89 | 35.65 | 35.57 | 36.17 |
| ga | 27.56 | 26.96 | 28.26 | 28.27 | mn | 33.39 | 33.91 | 34.63 | 30.04 | uk | 37.62 | 37.22 | 36.98 | 37.94 |
| gd | 27.95 | 27.52 | 28.69 | 28.92 | mr | 35.14 | 34.12 | 34.15 | 31.52 | ur | 34.82 | 34.30 | 34.10 | 29.01 |
| gl | 37.06 | 36.80 | 36.59 | 36.08 | ms | 35.25 | 35.04 | 35.01 | 30.61 | uz | 30.18 | 29.50 | 30.24 | 28.91 |
| gu | 28.25 | 27.83 | 29.09 | 31.06 | mt | 29.97 | 29.22 | 30.64 | 29.87 | vi | 38.13 | 37.67 | 37.60 | 38.31 |
| haw | 31.31 | 31.52 | 33.81 | 31.37 | my | 28.15 | 28.10 | 29.56 | 27.90 | yi | 29.22 | 29.01 | 29.49 | 27.86 |
| hi | 36.94 | 36.47 | 36.28 | 36.92 | ne | 34.37 | 33.53 | 34.34 | 32.72 | yo | 30.32 | 29.95 | 31.68 | 28.65 |
| hr | 37.25 | 36.94 | 36.93 | 28.00 | nl | 36.85 | 36.64 | 36.39 | 37.91 | zh | 38.85 | 39.26 | 39.50 | 40.30 |
| hu | 36.38 | 36.08 | 35.98 | 26.44 | no | 35.94 | 35.80 | 35.73 | 31.65 |  |  |  |  |  |
| hy | 33.36 | 32.66 | 32.98 | 26.73 | pl | 35.97 | 35.82 | 35.30 | 37.12 |  |  |  |  |  |

*Table 8.* **CLIP Score on the COCO2014 validation set.** 'Id' indicates the key of different language. SD15/21/XL (Rombach et al., 2022; Podell et al., 2023) represent our model implemented on the corresponding three T2I backbones, while AD refers to the AltDiffusion (Ye et al., 2023a). Our model also demonstrates strong text-to-image (T2I) capabilities across a wider range of languages.

and compared it with AltDiffusion (Ye et al., 2023a). More results are shown in Table 8. It can be observed that our model generalizes well to a wider range of languages and delivers impressive performance.

## B. More Qualitative Results

In this section, we present more examples of multilingual generation by the model, as well as examples of its interaction with existing community models and tools.

### B.1. Robustness to Multiple Languages

As shown in Figure 6, our model supports multiple languages, allows prompt inputs that combine different languages, and even recognizes emojis. For example, We can use *the car* emoji as a prompt (the first image in Figure 6), and the model can generate an image of a car.

### B.2. Plug and Play on Different Visual Generator

Our model can be seamlessly integrated into existing fine-tuned models, such as DreamShaper, Realistic Vision, and others. In Figure 6, we present additional examples, all generated by existing models interacting with Mulan. These examples use prompts in multiple languages, demonstrating support for various language combinations as inputs. Our model also supports some existing tools based on the Stable Diffusion series developed by the community. Here, we showcase several popular models, including LoRA (Hu et al., 2021) models, ControlNet (Zhang et al., 2023) models, and IP-Adapter (Ye et al., 2023b) models. The model's multilingual capabilities naturally come into play in these applications.

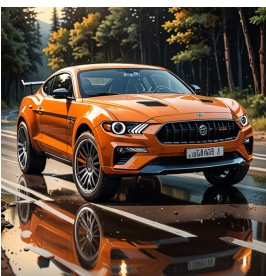 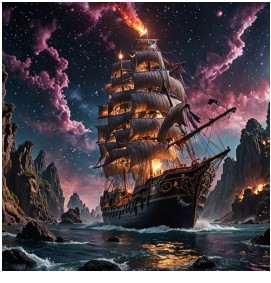 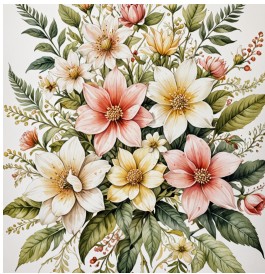 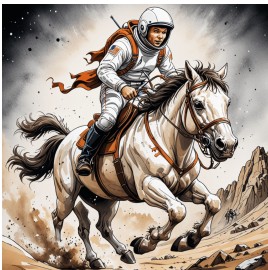

**Prompt:** 🚗
**Translation:** car

**Prompt**: Kozmikus máglyaködben rekedt kalózhajó
**Translation**:Pirate ship stranded in cosmic bonfire nebula

**Prompt**: Stunning botanical 水彩风格 白色背景
**Translation**: Stunning botanical, watercolor style, white background

**Prompt**: Astronauten rijden paarden in een schetsstijl
**Translation**: Astronauts riding horses in a sketch style

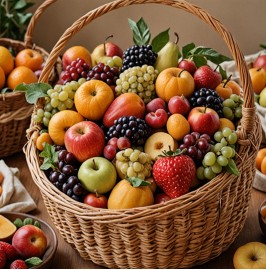 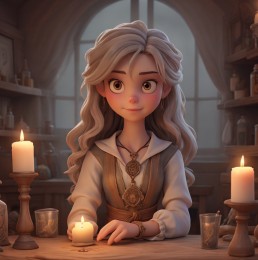 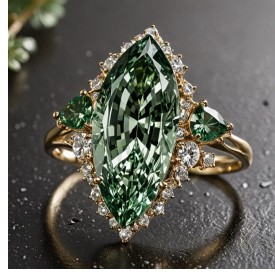 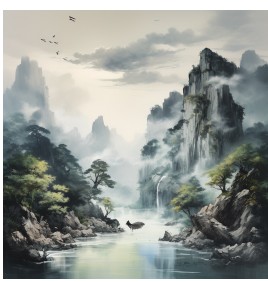

**Prompt**: Basket buahnya ada di meja
**Translation**: The fruit basket is on the table

**Prompt**: kadın büyücüsü
**Translation**: witch

**Prompt**: byzylyk me rruaza kristali
**Translation**: crystal bead bracelet

**Prompt:** 山水画
**Translation:** landscape painting

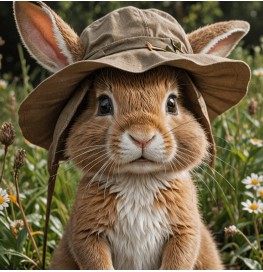 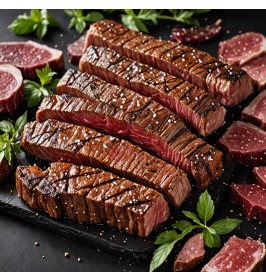 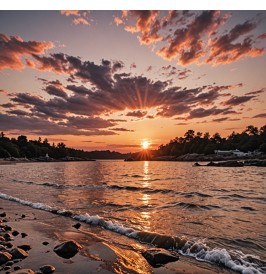 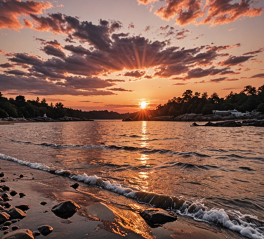

**Prompt**: 一只戴着帽子的 rabbit
**Translation**: A rabbit wearing a hat

**Prompt**: Thịt bò
**Translation**: Beef

**Prompt**: une seule photo d'un coucher de soleil sur la mer
**Translation**: a single photo of a sunset over the sea

**Prompt**: Ένας παπαγάλος που φοράει γυαλιά ηλίου
**Translation**: A parrot wearing sunglasses

*Figure 6.* **Multilingual generation results.** Our model supports multilingual and mixed-language inputs.

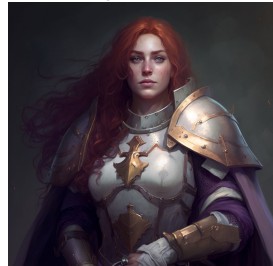 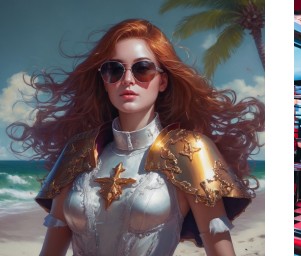 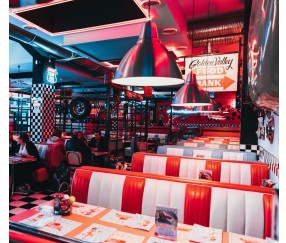 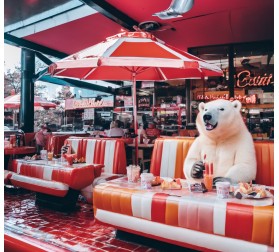

**Prompt:** ビーチでサングラスをかける
**Translation:** Wearing sunglasses on the beach

**Prompt:** 一只北极熊坐在椅子上喝着奶昔
**Translation:** a polar bear sitting in a chair drinking a milkshake

*Figure 7.* **IP-Adapter Results.** Our model enables multilingual style transfer by integrating with the IP-Adapter (Ye et al., 2023b).

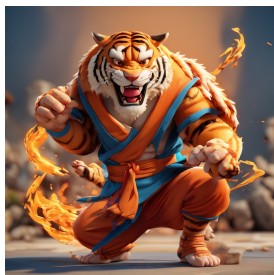 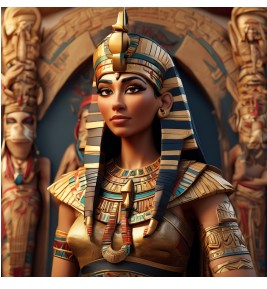 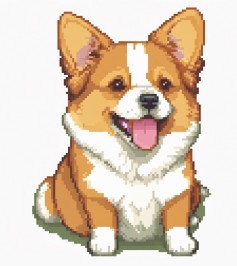 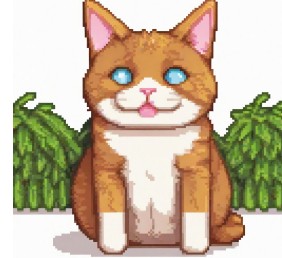

**Prompt:** 3d style kung fu tiger

**Prompt**: 3d style الملكة المصرية
**Translation**: 3d style Egyptian Queen

**Prompt**: 像素风格, 一只可爱的柯基
**Translation**: Pixel style-a cute corgi

**Prompt**: pixel 风格一只可爱的 🐱 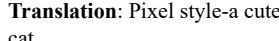
**Translation**: Pixel style-a cute cat

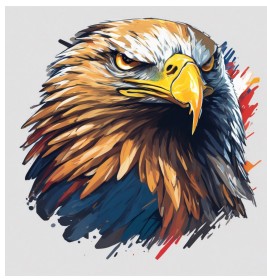 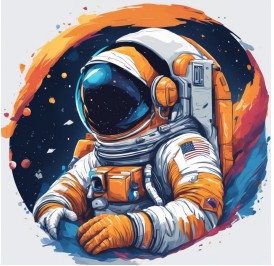 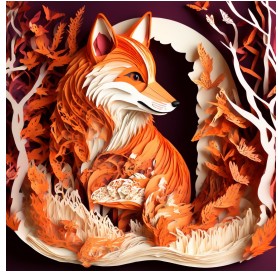 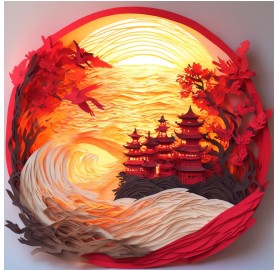

**Prompt**: vector art, 雄鹰
**Translation**: vector art, eagle

**Prompt**: vector art, 우주비행사
**Translation**: vector art, astronaut

**Prompt**: papercut-subject-キツネ
**Translation**: papercut-subject-fox

**Prompt**: papercut-subject-河流与落日
**Translation**: papercut-subject-river and sunset

*Figure 8.* **Lora Results.** Our model can naturally support multilingual input when using LoRA (Hu et al., 2021).

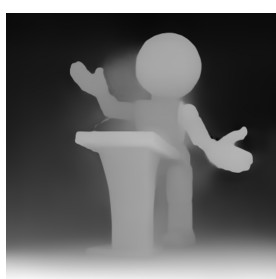 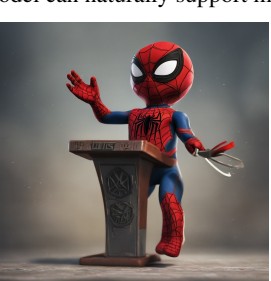 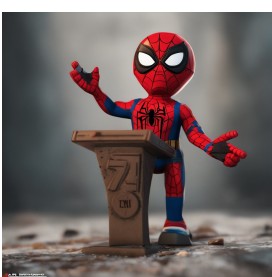 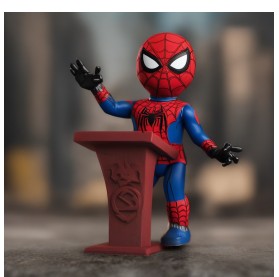

**Depth-ControlNet**

**Prompt**: 蜘蛛侠
**Translation:** spiderman

**Prompt**: 스파이더맨
**Translation:** spiderman

**Prompt**: スパイダーマン
**Translation**: spiderman

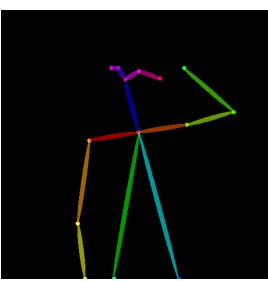 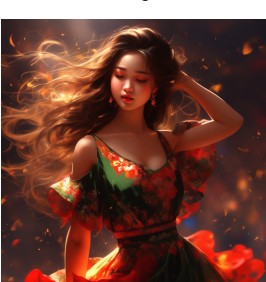 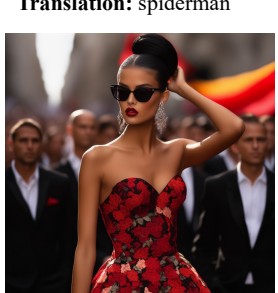 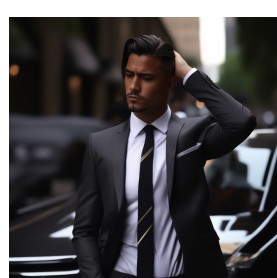

**OpenPose-ControlNet**

**Prompt**: 一个跳舞的女孩
**Translation**: A dancing girl

**Prompt**: Un mannequin international sur les podiums
**Translation**: An international model on the catwalks

**Prompt**: スーツを着たハンサムな男性
**Translation**: Handsome man in a suit

*Figure 9.* **ControlNet Results.** Our model can utilize existing ControlNet (Zhang et al., 2023) models, enabling multilingual image generation with conditional inputs such as depth maps and pose images.

