# OpenReview forum: "MuLan: Adapting Multilingual Diffusion Models for Hundreds of Languages with Negligible Cost"
_ICML.cc/2025/Conference — ICML 2025 poster_

### Official Review · Reviewer_fvJg · 2025-03-12

**Overall Recommendation:** 4

**Summary:**

This paper proposes MuLan, a text-encoder adapter that equips T2I models that are pre-trained with data dominant in English now with multilingual capabilities. With MuLan, T2I modes may take in prompts in purely non-English terms and
generate images in quality on par with those from English prompts.

**Claims And Evidence:**

**Issue #1: Lack of as-is baseline performances from English-only T2I backbone models.** Although the authors have shown comparisons with other multilingual T2I models such as AltDiffusion or GlueGen as in Table 3, what are the fundamental baseline performances that you simply feed in prompts in the target language as-is to an English-only T2I model, such as SD1.5? This as-is baseline setup is assumed to bring poor performance in ClipScore, certainly. But without showing these baselines, we may not have a solid idea how using a multilingual text encoder can make a significant difference in the first place.

As the authors have already implicated, the pre-training data used in the vanilla SD1.5 may have already had a language bias towards non-English Western languages. Thus, SD 1.5 should have higher as-is baselines in languages like Spanish or German. Will adding MuLan raise the performances across all languages by a universal increase, or will it instead selectively improve some languages than others over their as-is baselines? I don't think the robustness of the MuLan adapters can be truly demonstrated, if it is being presented without the as-is baselines.

**Essential References Not Discussed:**

References that are mentioned in my concerns in the sections above.

- [1] Improving Image Generation with Better Captions.  https://cdn.openai.com/papers/dall-e-3.pdf 2023

- [2] Text encoders bottleneck compositionality in contrastive vision-language models. https://arxiv.org/abs/2305.14897 EMNLP 2023

- [3] Evaluating Text-to-Visual Generation with Image-to-Text Generation. https://arxiv.org/pdf/2404.01291 ECCV 2024

**Experimental Designs Or Analyses:**

Please refer to my concerns in the Claims and the Method sections above.

**Methods And Evaluation Criteria:**

**Issue #2: Confusion over the ClipScore specification.** According to Line 288, the CLIPScore/SIM used in the paper uses InternVL-LLaMA as the surrogate model. However, CLIPScore is mostly widely calculated with CLIP-VIT variants, such as in Dall-E 3 [1]. How do the MuLan adapters perform if measured in industry standard metrics?

**Issue #3: T2I Metrics Other than ClipScore?** CLIPScore has already been proven to struggle at compositional text prompts as in [2]. Since the prompts in XM12 mostly feature compositional attributive phrases like those in Figure 1, how well does MuLan improve then, if measured in fine-grained text-image-alignment metrics such as VQAScore [3]?

**Other Comments Or Suggestions:**

None

**Other Strengths And Weaknesses:**

None.

**Questions For Authors:**

Please find my 3 major issues in the sections above. Out of all, Issue #1 has the highest severity and will greatly affect my impression of this work if left unaddressed.

After all, I really like the potential application of MuLan, but I would like to push its presentation over the high-standard bar of ICML. I am open to updates and would like to engage with the author further.

**Relation To Broader Scientific Literature:**

The potential of MuLan can be high, as it works in a plug-and-play / training-free style that can be easily integrated into established T2I pipelines, such as those implemented in the Huggingface framework.

**Theoretical Claims:**

There is no major deviation from mainstream theories.

---

> ### Author Rebuttal · Authors · 2025-03-31
>
> Dear reviewer fvJg,
>
> Thanks so much for your constructive comments and support for acceptance. We hope our responses can address your concerns.
>
> **Q1: Lack of as-is baseline performances from English-only T2I backbone models.**
>
> **A1**: We thank the reviewer for emphasizing the importance of “as-is” baseline performances. In response, we evaluated Stable Diffusion v1.5 and PixArt-α by directly inputting prompts in 11 languages from the XM12 dataset, using InternVL-LLaMA to compute CLIP scores. These as-is results were then compared with our MuLan-adapted models.
>
> We found that **MuLan adapters consistently improved performance across all languages.** Notably, even for languages close to English—such as French, Spanish, and German—our method still achieved clear gains over the as-is baseline, demonstrating its broad effectiveness.
>
> |Model|avg|de|fr|it|es|pl|hi|ru|zh|ja|ko|ar|
> |-|-|-|-|-|-|-|-|-|-|-|-|-|
> |SD15 (as-is)|28.1|31.9|34.2|31.8|34.2|24.8|26.8|23.0|26.6|26.9|24.6|24.5|
> |MuLan-SD15|**37.2**|38.0|38.0|37.8|37.7|37.0|35.6|37.4|36.6|36.7|36.4|38.2|
> |Δ (SD15)|**+9.1**|+6.1|+3.8|+6.0|+3.5|+12.2|+8.8|+14.4|+10.0|+9.8|+11.8|+13.7|
> |PixArt(as-is)|29.0|36.8|38.2|36.0|36.9|27.0|24.0|28.9|24.2|22.1|22.4|22.2|
> |MuLan-PixArt|**39.5**|40.5|40.2|40.0|39.6|39.1|37.2|39.6|39.3|40.5|39.3|39.1|
> |Δ(PixArt)|**+10.5**|+3.7|+2.0|+4.0|+2.7|+12.1|+13.2|+10.7|+14.9|+18.4|+16.9|+16.9|
>
> **Q2: Confusion over the CLIPScore specification.**
>
> **A2:** While the industry-standard CLIP-ViT models are indeed widely used, they are primarily trained on English data and thus provide unreliable similarity scores for non-English inputs. In contrast, InternVL-LLaMA has been trained on multilingual image-text pairs and possesses better understanding of non-English languages. Therefore, we chose InternVL-LLaMA as the surrogate model for computing CLIPScore in our paper.
>
> To further ensure the objectivity of our CLIPScore evaluation, we additionally used the multilingual CLIP-ViT model released by LAION to compute similarity scores. The results, shown in the table below, demonstrate that **our model still performs strongly and that the observed trends are consistent with those reported in Table 3 of the paper.**
>
> |model|avg|en|fr|es|it|zh|ja|hi|de|ko|ru|ar|pl|
> |-|-|-|-|-|-|-|-|-|-|-|-|-|-|
> |GlueGen|21.05|22.3|21.3|20.6|19.6|21.5|21.0|-|-|-|-|-|-|
> |AltDiffusion|23.13|24.2|23.6|23.2|23.1|24.3|23.0|21.1|24.3|22.5|22.3|22.7|23.2|
> |SD15(Google Translate)|22.27|22.3|23.8|23.4|22.8|22.6|21.3|20.8|24.5|21.7|22.6|19.5|21.9|
> |PixArt(Google Translate)|24.26|24.1|24.2|23.8|24.5|26.4|25.1|21.6|26.5|23.1|24.6|23.3|23.9|
> |MuLan-SD15|23.02|21.8|23.6|23.2|22.7|23.6|24.2|22.7|24.5|21.8|23|22.3|22.9|
> |MuLan-PixArt|24.15|24.4|23.9|23.4|24.2|25.7|24.8|23.2|25.8|23.4|24.6|23.1|23.4|
>
> **Q3: Limitations of CLIPScore.**
>
> **A3:** The prompts in XM12 predominantly feature compositional text prompts. While CLIPScore is useful for evaluating whether the main subject in the image aligns with the prompt, it falls short in assessing object-level details and spatial relationships. As you suggested, we additionally evaluated our model using VQAScore to better measure fine-grained text-image alignment.
>
> Since the default VQAScore evaluation model (clip-flant5-xxl) does not support multilingual prompts adequately, we adopted GPT-4o as our evaluation model to enable multilingual VQAScore assessment. Results show that our model maintains strong performance, significantly outperforming GlueGen and AltDiffusion, and even surpassing or matching translation-based baselines across most languages.
>
> Our model effectively leverages the capabilities of existing MLLMs, demonstrating strong generalization in multilingual image generation. In the future, we plan to further integrate the native multilingual, contextual, and reasoning abilities of MLLMs into the image generation process.
> |model|avg|en|fr|es|it|zh|ja|hi|de|ko|ru|ar|pl|
> |-|-|-|-|-|-|-|-|-|-|-|-|-|-|
> |GlueGen|0.533|0.81|0.51|0.52|0.45|0.41|0.5|-|-|-|-|-|-|
> |AltDiffusion|0.581|0.71|0.6|0.6|0.59|0.5|0.53|0.62|0.49|0.56|0.51|0.64|0.62|
> |SD15(Google Translate)|0.612|0.81|0.66|0.71|0.61|0.57|0.5|0.67|0.5|0.7|0.68|0.41|0.52|
> |PixArt(Google Translate)|0.750|0.85|0.78|0.8|0.82|0.71|0.71|0.82|0.71|0.76|0.76|0.59|0.69|
> |MuLan-SD15|0.635|0.80|0.74|0.76|0.68|0.60|0.52|0.53|0.57|0.65|0.75|0.45|0.57|
> |MuLan-PixArt|0.744|0.88|0.81|0.83|0.81|0.71|0.71|0.61|0.73|0.74|0.79|0.59|0.72|
>
> References
>
> [a1] https://huggingface.co/laion/CLIP-ViT-H-14-frozen-xlm-roberta-large-laion5B-s13B-b90k
>
> [a2] Evaluating Text-to-Visual Generation with Image-to-Text Generation. https://arxiv.org/pdf/2404.01291 ECCV 2024

---

> > ### Comment · Reviewer_fvJg · 2025-04-04
> >
> > Thank you so much in the feedback. I believe my concerns have been adequately addressed. I am happy to update my rating. I am looking forward to seeing if MuLan can be applied to more sophisticated backbones (e.g. Flux.1) in the future since SD 1.5 has already become so obsolete as we speak.

---

> > > ### Author Response · Authors · 2025-04-09
> > >
> > > Thank you for raising the score and for your constructive suggestions! We’re glad our responses addressed your concerns, and we will incorporate the additional discussions and results into the paper. We will also try to apply Mulan to more advanced text-to-image models in the future.

---

### Official Review · Reviewer_RQVZ · 2025-03-14

**Overall Recommendation:** 3

**Summary:**

This paper proposes a simple yet effective way to handle multilingual text input in text-to-image generation. By utilizing a pre-trained multilingual text encoder and introducing a light-weighted adapter, the resulting model is shown to handle multilingual input well.

**Claims And Evidence:**

The paper claims to propose a method which can handle multilingual text input well in text-to-image generation. According to the results in the paper, it obtains good results than some previous methods.

**Essential References Not Discussed:**

None

**Experimental Designs Or Analyses:**

The experimental design seems reasonable. The authors compared the proposed method with related methods, and also included some important baseline such as directly using Google translate to process multilingual input.

**Methods And Evaluation Criteria:**

The idea of utilizing a pre-trained multilingual text encoder is reasonable, because it has already been pre-trained on multilingual text and should be able to better align different languages into a single embedding space.

**Other Comments Or Suggestions:**

None

**Other Strengths And Weaknesses:**

According to the results in Table 3. The simple baseline, which directly use Google translate to process multilingual input obtains comparable results with the proposed method:  SD 1.5 baseline outperforms the proposed SD1.5 version model on 5/12 cases, and PixArt baseline outperform corresponding proposed method on 8/12 cases. People may choose to use baseline method because it doesn't require any training, and can choose arbitrary translation models and T2I models. Meanwhile, the proposed method has to be trained again if one want to apply it to a new SoTA model.

**Questions For Authors:**

What is the reason causing the performance difference between different alignment methods in Table 5?

**Relation To Broader Scientific Literature:**

The difference is that the paper utilizes a pre-trained multilingual text encoder, while previous methods either fine-tuned the diffusion model on multilingual text-image dataset (may lead to image quality drop and potential bias) or align different languages with light-weighted network.

**Theoretical Claims:**

There is no theoretical claims in the paper.

---

> ### Author Rebuttal · Authors · 2025-04-01
>
> Dear reviewer RQVZ,
>
> Thanks very much for your valuable comments. We hope our responses can address your concerns and clarify our contribution.
>
> **Q1. Comparison with translation baseline.**
>
> **A1:** In fact, previous multilingual text-to-image generation works (e.g., AltDiffusion) typically compare against open-source translation models such as NLLB [a1]. As shown in the following table, we also compared the results obtained using NLLB as a translation tool, and in 11 non-English languages of XM12, **our performance exceeded the baseline of NLLB as a translation tool**. We can also observe that using translation tools to handle non-English input is highly dependent on the performance of the translation tool, being very sensitive to its performance.
> |Model|open/closed source|avg|fr|es|it|zh|ja|hi|de|ko|ru|ar|pl|
> |-|-|-|-|-|-|-|-|-|-|-|-|-|-|
> |SD15 (NLLB)|open|36.17|37.6|37.4|37.2|36.2|36.2|32.7|37.7|35.9|36.8|33.9|36.3|
> |SD15 (Google Translate)|closed|36.70|38.2|38.0|37.8|36.6|36.7|33.1|38.4|36.4|37.4|34.4|36.7|
> |MuLan-SD15|open|**37.60**|38.0|37.7|37.8|37.7|38.0|35.6|38.0|37.6|37.9|38.2|37.1|
> |PixArt (NLLB)|open|38.27|39.7|39.4|39.2|38.1|38.6|34.4|40.3|37.7|39.3|35.6|38.7|
> |PixArt (Google Translate)|closed|**39.82**|41.2|40.6|41.0|39.9|40.8|34.7|41.5|39.0|40.5|39.0|39.9|
> |MuLan-PixArt|open|39.49|40.2|39.6|40.0|39.3|40.5|37.2|40.5|39.3|39.6|39.1|39.1|
>
> **Google Translate** is a strong commercial system with high development and data costs, and has undergone long-term refinement to reach its current performance, **it remains a closed-source tool, making it unsuitable for offline or privacy-sensitive scenarios**.
>
> In contrast, **MuLan is a low-cost baseline that achieves multilingual image generation by training on English image-text pairs and is entirely based on open-source suites**. In terms of the **FID** metric, our model generally **achieves better results**. On CLIPScore, our results are already comparable to those obtained using Google Translate, and we show clear advantages in less common languages. These results highlight the potential of achieving multilingual image generation through training, without relying on external translation systems. We will include these points in the main paper in future revisions.
>
> **Q2. Adapt to the new SoTA model.**
>
> **A2:** One of our main contributions is the low adaptation cost (e.g., **MuLan-SD15 training only requires 96 GPU hours**). As demonstrated by the visual results in Figure 1 and Figure 4, the Adapter trained for Mulan-SD15 can be effectively applied to other fine-tuned variants of SD15 (e.g., Dreamshaper), and also works well with plugins such as LoRA and ControlNet. The same holds for models like SD21, SDXL, and PixArt-$\alpha$. This indicates that our method exhibits strong generalizability across this family of models, with minimal additional cost. For newly emerging SoTA models, our method can still be employed to support them and their related plugins and variants with similarly low adaptation cost.
>
> **Q3. More analysis of performance differences in table 5**
>
> **A3:** Based on the experimental results in Table 5, we can draw two main conclusions.
>
> **1. Only the aligned text encoder can support multilingual image generation through MuLan.**
>
> By comparing the models in Rows 1–3 and Rows 4–8 of Table 5, it is evident that without Language-Centered Alignment (LC) or Image-Centered Alignment (IC), the models are unable to support multilingual image generation through MuLan. We provide visualizations of the feature distributions for some of these models in Figure 5 of the appendix. As shown, models without alignment—such as LLaMA2-7B (Figure 5(d)) and XLM-R Large (Figure 5(a))—produce scattered features when processing inputs with the same meaning but in different languages. In contrast, aligned models—such as XLM-R Large* (Figure 5(b)), Mul-OpenCLIP (Figure 5(c)), and InternVL-LLaMA (Figure 5(e))—can cluster features of semantically equivalent inputs across different languages. This is a key reason why our method, trained solely on English image-text data, is still able to support multilingual image generation.
>
> **2. Image-Centered Alignment outperforms Language-Centered Alignment.**
>
> By comparing the models in Rows 4–6 and Rows 7–8, we observe that LC (Language-Centered Alignment) yields inferior results compared to IC (Image-Centered Alignment). This is primarily because LC relies on translated data, which may introduce noise due to inaccuracies in translation. Moreover, IC aligns multilingual semantic features through the shared image feature space, effectively providing MuLan with a prior that facilitates further refinement of the alignment. In contrast, LC relies entirely on MuLan to establish the connection between language features and image features from scratch. These factors collectively contribute to the inferior performance of LC compared to IC.
>
> References
>
> [a1] https://huggingface.co/facebook/nllb-200-3.3B

---

### Official Review · Reviewer_BvZv · 2025-03-14

**Overall Recommendation:** 4

**Summary:**

This paper introduces MuLan, a lightweight and plug and play language adapter that enables multilingual text to image generation for diffusion models with minimal computational cost. The central idea is that multilingual text encoders can be used to enable multilingual image generation without the need for extensive labeled datasets in multiple languages. Instead of training a full diffusion model on multilingual text-image pairs, the model freezes the text encoder and diffusion model and introduces a small adapter module that bridges the two. This approach allows model to support over 110 languages while requiring training only on English-language image-text pairs.

The main claims are that the model achieves strong multilingual image generation performance (on par with models explicitly trained on multilingual text image pairs) and that it requires significantly less computational cost and training data (by using leveraging pretrained multilingual text encoders and training only a small adapter) and that it Is flexible and integrates with existing models and community tools (e.g. LoRA, Control Net).

The authors evaluate the model on multilingual benchmarks and compare it against other approaches from the literature and translation-based approaches. The results show strong performance in both high-resource and low-resource languages, with significant efficiency gains in terms of compute and data requirements, suggesting that the proposed model is both computationally efficient and broadly applicable.

**Claims And Evidence:**

The paper makes several strong claims about the model. Overall, the claims seem well supported by the presented evidence.

First, they claim that the proposed model achieves multilingual performance comparable to English trained models while requiring only English data. This is supported by quantitative results, where the CLIP similarity score for English (39.57) is nearly identical to the average score across all other languages (39.61). The authors also show visual examples demonstrating the model's ability to generate accurate images from prompts in diverse languages, including low resource languages. Second, they claim that in comparison to other multilingual diffusion models, the model reduces training costs. They argue that because the model only trains a small adapter, it avoids the high cost of training a full diffusion model on multilingual datasets. The claim is backed by the reported training time being orders of magnitude lower than the competing models that MuLan is benchmarked against, and by a comparative cost analysis against. Another claim is that the model generalizes across a large number of languages, which is supported by an evaluation on multilingual datasets showing that the model is comparable to or sometimes better than translation-based methods.

**Essential References Not Discussed:**

I would have expected acknowledgement of works like MAD-X [1] for adapter based cross lingual transfer. Other than that, several relevant works are cited (and of course directly compared against) and it is acknowledged that this work touches several disciplines.

[1] https://arxiv.org/abs/2005.00052v2

**Experimental Designs Or Analyses:**

As discussed, the methodology and evaluation of this paper are strong. The methodology for language and image centered alignment is sound and aligns with common practice in the representation learning literature. The evaluation and benchmarking provide a broad test of generalization across low and high resource languages. The translation based baselines compare whether the direct multilingual generation is better than translation pipelines. The analysis of computational efficiency is thorough and well documented, illustrating very clearly the contrast with other multilingual models in the literature. The comparison of image vs language centred alignment also seems sound and empirically supports the claim that image centred alignment is the superior choice.

As before, a minor remark would be the lack of a cross lingual consistency evaluation. One can imagine that when testing across so many different languages, stylistic/cultural nuances could affect the results, and there could be some failure cases. Finally, as another minor remark, there is information about experimental design to aid reproducibility but it is not clear how some of the hyperparameter settings were arrived at or whether they are just default values.

Overall there are no real comments from me on this, and methodologically and experimentally this paper is quite strong and comprehensive.

**Methods And Evaluation Criteria:**

The method essentially involves training a multilingual adapter on top of diffusion models. The diffusion model itself is frozen, and the adapter is trained to connect it to a pretrained multilingual text encoder. This is quite appropriate for the context and enables text to image generation from a wide variety of languages while keeping the underlying generation model fixed with savings in terms of required compute. The adapter learns to map multilingual text embeddings into the same space as the english trained diffusion model. The paper explores 2 different strategies to achieve this: language centred alignment (aligning multilingual embeddings to English using parallel translation datasets and a distillation loss) and image centred alignment (using contrastive learning to align multilingual embeddings with image embeddings, ensuring the text prompts in different languages produce similar image representations). The experiments show that the image centred approach performs better - likely because the alignment is done directly between language and image space as opposed to relying on noisier translation datasets.

The evaluation metrics are the CLIP similarity score (text to image alignment) and FID score which measures image quality. These metrics are quite appropriate in this context and (especially in the case of FID) widely used across the literature. Furthermore, the model is benchmarked on XM3600 (12 languages) and COCO2014 (85 languages) and compared against a number of strong baselines (e.g. translation based stable diffusion) which strengthen the claims made by the authors.

A minor remark here: it would be interesting to have a more detailed evaluation of cross lingual consistency e.g. whether a given concept is represented similarly across these languages and the degree to which similar results are obtained. It is understood however, that this is rather challenging and perhaps wouldn't strengthen the claims of this paper significantly enough.

**Other Comments Or Suggestions:**

not applicable

**Other Strengths And Weaknesses:**

The paper is well written and clearly presented. The methods are not very novel, but they are timely and of considerable impact. Nothing further to report beyond what was already discussed.

**Questions For Authors:**

not applicable

**Relation To Broader Scientific Literature:**

The papers's key contributions align with several existing research areas in NLP and image generation. Adapter based methods for multilingual NLP are quite are a widely explored concept to enable multilingual capability without retraining whole models. The paper under discussion extends the idea to text to image generation. Indeed, due to the extensive commercial applications, text to image models have also been quite a popular area of research recently, so this paper is very timely in both respects.

**Theoretical Claims:**

not applicable.

---

> ### Author Rebuttal · Authors · 2025-04-01
>
> Dear reviewer BvZv,
>
> Thanks a lot for your insightful reviews and support for our work! We hope our responses can address your questions.
>
> **Q1. Lack of a cross lingual consistency evaluation**
>
> **A1:** We appreciate the reviewer’s suggestion to evaluate cross-lingual conceptual consistency. Our work focuses on enabling multilingual text-to-image generation by leveraging a pretrained multilingual text encoder and English image-text pairs. While our current method demonstrates some degree of cross-lingual consistency, it may struggle with culture-specific corner cases, which fall outside the scope of this work and would require human curation or high-quality data to resolve. We will acknowledge this limitation in the main paper and plan to address it in future work. As a potential direction, we may explore collecting data with explicit culture-specific concepts and developing a dedicated benchmark for evaluating cross-lingual consistency in multilingual T2I generation.
>
> **Q2. Clarification on hyperparameter settings**
>
> **A2:**
> The training hyperparameters for MuLan-SD15 and MuLan-PixArt used in Table 3 are already provided in Section 4.1. Here we provide additional details for completeness: we use the AdamW optimizer with $\beta = (0.9, 0.999)$ and a weight decay of 0.01. For SD 1.5, SD 2.1, and PixArt, we follow the original models’ resolutions (512x512 or 768x768). For SDXL, we adopt a two-stage training strategy with different resolutions to ensure stability. We adopt classifier-free guidance by randomly dropping text conditions, following [a1].
>
> In Table 5, we only replace the language model; all other training settings are identical to those of MuLan-SD15.
>
> We will include these details regarding the selection of experimental hyperparameters in the updated manuscript.
>
> **Q3. Ablation study on the model size**
>
> **A3:**
>
> **For the size of the adapter**
>
> In the main paper, our proposed adapter uses a lightweight one-layer Transformer encoder-decoder structure (see Section 3.2). To further investigate the relationship between model size and performance, we have conducted additional ablation experiments on Stable Diffusion 1.5, testing adapters with 2, 4, and 6 Transformer layers. The experimental setup follows Appendix A.1. We calculated the average CLIPScore across seven languages from the XM12 dataset.
>
> |layers|1|2|4|6|
> |-|-|-|-|-|
> |**Avg CLIPScore**|35.8|35.8|35.5|35.4|
>
> We observe that increasing the number of Transformer layers in the adapter does not lead to improved performance; in fact, it **slightly degrades performance**. This result aligns with previous findings in works such as LLaMA-Adapter [a2] and MiniGPT-4 [a3], which demonstrate **that adapter-based tuning for LLMs typically does not require large parameter counts or massive training data**. Our lightweight adapter design is consistent with these community practices, suggesting that a compact architecture is sufficient and may even be more stable and efficient for adaptation.
>
> **For the size of the language model**
>
> Our approach relies on a series of pretrained multilingual language models, making it difficult to perform strictly controlled ablation studies on the size of the language model. These models differ not only in parameter count but also in architecture, tokenizer, training corpora, which complicate direct comparisons.
>
> Although a strictly controlled study is not feasible, a coarse-grained observation from Table 5 reveals a trend suggesting that the size of the language model may impact the performance upper bound. Specifically, the first four models—MultiLang-CLIP (33.2), AltClip-m18 (33.3), XLM-R Large* (34.7), and Mul-OpenCLIP (36.1)—all use **XLM-R Large(335M)** as the language model. In contrast, **InternVL-LLaMA(7B)**, which achieves the highest score of **37.8**.
>
> Since our approach builds upon existing pretrained multilingual models, conducting strictly controlled comparisons on model size is non-trivial. As part of future work, we plan to explore how our adapter performs when integrated with increasingly stronger language models, which could further reveal its potential in real-world multilingual generation.
>
> References
>
> [a1] Ho, Jonathan. “Classifier-Free Diffusion Guidance.” https://arxiv.org/abs/2207.12598
>
> [a2] Zhang, Renrui et al. “LLaMA-Adapter: Efficient Fine-tuning of Language Models with Zero-init Attention.” https://arxiv.org/abs/2303.16199.
>
> [a3] Zhu, Deyao et al. “MiniGPT-4: Enhancing Vision-Language Understanding with Advanced Large Language Models.” https://arxiv.org/abs/2304.10592

---

> > ### Comment · Reviewer_BvZv · 2025-04-09
> >
> > I thank the authors for their insightful reply. In light of it, I am happy to keep the current score and for the paper to be accepted at ICML.

---

> > > ### Author Response · Authors · 2025-04-09
> > >
> > > Thank you for your positive feedback and endorsement; we will continue refining the paper to present our best work at ICML.

---

### Decision · Program_Chairs · 2025-05-01

**Decision:**

Accept (poster)

**Comment:**

This paper focuses on multilingual text-to-image generation and introduces a lightweight adapter to enable that on diffusion-based models with minimal computational cost. During training, only the adapter is optimized, while the text encoder and diffusion model remain fixed. This approach supports over 100 languages, although only English-image pairs are used for training. Reviewers appreciate the comprehensive experiments and efficiency improvements demonstrated in the paper. AC agrees with the reviewers in recommending acceptance and encourages the authors to incorporate the authors-reviewers discussions into the final paper and apply the method to more recent image generation models (*e.g.*, Flux).